# Resistance to Antibody-Drug Conjugates Targeting HER2 in Breast Cancer: Molecular Landscape and Future Challenges

**DOI:** 10.3390/cancers15041130

**Published:** 2023-02-10

**Authors:** Lorenzo Guidi, Gloria Pellizzari, Paolo Tarantino, Carmine Valenza, Giuseppe Curigliano

**Affiliations:** 1Division of New Drugs and Early Drug Development for Innovative Therapies, European Institute of Oncology, IRCCS, 20139 Milan, Italy; 2Department of Oncology and Haemato-Oncology, University of Milan, 20122 Milan, Italy; 3Department of Medical Oncology, Dana-Farber Cancer Institute, Boston, MA 02215, USA; 4Breast Oncology Program, Harvard Medical School, Boston, MA 02215, USA

**Keywords:** HER2-positive breast cancer, antibody-drug conjugates, trastuzumab emtansine, trastuzumab deruxtecan, mechanisms of resistance

## Abstract

**Simple Summary:**

Human epidermal growth factor receptor 2 (HER2)-positive metastatic breast cancer (mBC) represents a subgroup of breast cancer characterized by an aggressive behaviour and a particular sensitiveness to HER2-targeted agents. Because of the frequent occurrence of acquired resistance to anti-HER2 therapy, novel agents were investigated, including antibody-drug conjugates (ADCs). ADCs represent an emerging class of anticancer agents consisting of a humanized monoclonal antibody bounded to a cytotoxic drug by a molecular linker. HER2-positive mBC represents the first solid tumor in which ADCs, like Trastuzumab emtansine (T-DM1) and Trastuzumab deruxtecan (T-DXd), demonstrated to improve clinical outcomes compared to prior standards of care. Nonetheless, despite their effectiveness, resistance to T-DM1 and T-DXd still occurs in most of the patients treated with these ADCs, warranting an improved understanding of the mechanisms underlying resistance. This review aims to describe the emerging mechanisms of resistance to ADCs targeting HER2, highlighting the potential strategies to overcome resistance and further improve clinical outcomes of patients with metastatic breast cancer.

**Abstract:**

The treatment of HER2-positive metastatic breast cancer (mBC) with Trastuzumab emtansine (T-DM1) and Trastuzumab deruxtecan (T-DXd), two antibody-drug conjugates (ADCs) targeting HER2, is burdened by progression of disease related to the acquisition of mechanisms of resistance. Resistance to T-DM1 is caused by the decrease of HER2 expression, the alteration of intracellular trafficking, the impairment of lysosome functions, the drug expulsion through efflux pumps and the activation of alternative signal pathways. Instead, the decrease of HER2 expression and *SLX4* loss of function mutations represent the first evidences of mechanisms of resistance to T-DXd, according to the results of DAISY trial. Several strategies are under evaluation to overcome resistances to anti-HER2 ADCs and improve clinical outcomes in patients progressing on these agents: combinations with tyrosine kinase inhibitors, statins, immune checkpoint inhibitors and synthetic DNA-damaging agents are emerging as promising approaches. Furthermore, novel anti-HER2 ADCs with innovative structures and mechanisms of action are in development, in the attempt to further improve the activity and tolerability of currently available agents.

## 1. Introduction

Breast cancer (BC) represents the leading cancer for incidence and mortality in women, with 287,000 estimated new cases and 43,000 deaths in 2022 in United States [1,2]. Human epidermal growth factor receptor 2 (HER2)-positive BC accounts for approximately 15–20% of BCs and is associated with an aggressive phenotype, high recurrence rates and inferior survival outcomes, if left untreated [3].

The poor prognosis of this BC subtype was significantly improved with the advent of monoclonal antibodies (mAb) targeting HER2, namely trastuzumab and pertuzumab, and small molecules like tyrosine kinase inhibitors (TKI), such as lapatinib, tucatinib and neratinib. These agents demonstrated a clinical benefit in terms of progression/disease free survival (PFS/DFS) and overall survival (OS). However, most of patients with HER2-positive metastatic BC (mBC) develop resistance to HER2-directed therapies and experience progression of disease (PD) [3,4]. 

In order to delay the occurrence of resistance generation and to further improve clinical outcomes, new agents incorporating the HER2-targeted antitumor properties of trastuzumab with the cytotoxic activity of chemotherapy were developed, namely antibody-drug conjugates (ADCs) [5].

ADCs are an evolving class of anticancer drugs consisting of a humanized mAb binding a tumor associated-antigen, conjugated to a cytotoxic drug (payload) by a molecular linker; the canonical model of ADC action posits the following: binding of the mAb to the target antigen, subsequent internalization and, finally, linker breakdown and intracellular payload release. ADCs can be conceived of as prodrugs, which in many cases require processing and metabolism by the target cells before their end activity can be fully realized. Upon administration, the ADC formulation contains three major circulating components: the conjugate (which constitutes the overwhelming fraction), naked antibodies and free payload molecules [6].

Efficacy of ADCs depends on each of these elements and their specific features [7]. 

mAbs incorporated in ADCs are mainly based on the igG1 isotype because of its higher immunogenic properties and easier production if compared to the other igG2 and igG4 subtypes [8]; the drug to antibody ratio (DAR) is usually between 2 and 8. 

Linkers affect all pharmacokinetic properties of ADCs and can be cleavable or noncleavable; non-cleavable linkers are more stable and are cleaved only by the complete proteolytic degradation of the ADC in lysosomes [9]. Instead, cleavable linkers release the payload in response to tumor-associated factors, such as acid pH (acid-labile linkers), reduction-oxidation conditions (disulfide-linkers), abundance of proteolytic enzymes (protease-cleavable linkers), allowing the diffusion of the payload through neighbouring cells not expressing the target (bystander effect) [10].

Lastly, payloads derive from DNA-damaging agents (duocarmazine), antimicrotubule compounds (mertansine, vedotin, amberstatin) or topoisomerase inhibitors (deruxtecan, govitecan), and are characterized by an extremely unfavourable therapeutic index, if administered systemically as free drugs [8].

Trastuzumab emtansine (T-DM1) was the first ADC approved in solid tumors in 2013 [11], in particular, in patients with HER2-positive mBC [12], according to the OS benefit demonstrated in a clinical trial enrolling patients progressing on trastuzumab and chemotherapy; it is an ADC targeting HER2, conjugated through a non-cleavable linker to an anti-microtubule compound (mertansine or DM1), with a DAR of 3.5 and it is characterized by a multiple mechanism of action, from the selective deliver of the payload to HER2 signalling inhibition and antibody dependent cell-mediated cytotoxicity (ADCC) mediated by trastuzumab [13,14].

By that time, the progressive transition from non-cleavable to cleavable linkers, the optimization of Abs structures and the improvement of DAR, leaded to the development of new generations of anti-HER2 ADCs in mBC, among whom Trastuzumab deruxtecan (T-DXd) first received the Food and Drug Administration (FDA) and European Medicines Agency (EMA) approval [15]. T-DXd is an anti-HER2 ADC characterized by a DAR of 8, a cleavable linker and a topoisomerase inhibitor (deruxtecan) as payload; it demonstrated to provide an OS benefit in patients with mBC both HER2-positive and HER2low, which is defined by HER2 immunohistochemistry (IHC) score of 1+ or 2+ with absence of HER2 amplification at in situ hybridization (ISH) test [16,17,18].

To date, many other clinical trials are testing novel anti-HER2 ADCs (Figure 1). They include: Trastuzumab Duocarmazine (SYD985), which is formed by the conjugation of trastuzumab to the alkylating agent duocarmycin mediated by a cleavable linker with a DAR of 2.8:1; Disitamab Vedotin (RC48-ADC), 101 which is characterised by the conjugation of the anti-HER2 mAb disitamab to the synthetic antineoplastic agent monomethyl auristatin E (MMAE), an anti-microtubule agent, with a DAR of 4; XMT-1522 (TAK-522), a new generation ADC targeting different HER2 epitopes than trastuzumab, whose development was recently halted due to safety issues; ARX788, a next-generation site-specific anti-HER2 which is conjugated with amberstatine, an anti-microtubule agent, among others [7,19,20,21,22].

However, despite this revolution in the treatment of HER2-positive mBC, patients eventually experience PD also under ADCs, due to the occurrence of acquired mechanisms of resistance to the mAb, the payload or both [23]. The molecular landscape of resistance mechanisms to these agents is very heterogeneous and has not still been fully depicted; in fact, T-DXd was granted by FDA for accelerated approval on December 2019 [24].

This narrative review aims at summarizing the known and emerging resistance mechanisms to anti-HER2 ADCs, and at highlighting the most promising approaches to overcome resistance.

## 2. Mechanisms of Resistance to T-DM1

The mechanism of action of T-DM1 involves both the IgG1 backbone, which induces antibody-dependent cellular cytotoxicity (ADCC) through the fragment crystallisable (Fc) and PI3K/AKT signalling inhibition by HER2 binding with the fragment antigen-binding (Fab), and the mertansine, a microtubule disrupting agent which is released in the cytosol, where determines cell cycle arrest in G2/M phases and cell death [25].

Therefore, the mechanisms of resistance to T-DM1 can be attributed to the antibody or the payload, and can be shared with trastuzumab or anti-microtubules agents (e.g., taxanes) [14]; however, the evidences of activity and efficacy to T-DM1 in patients progressing on trastuzumab and taxanes highlights that ADCs cannot be reduced to the mere sum of their components.

Mechanisms of resistance to T-DM1 can be classified as receptor-related, highlighting the importance of target accessibility, or intracellular, underlying the pivotal role of each process after the internalization, namely the presence of drug efflux pumps and the complex regulation of the intracellular signalling pathways [14]. 

Furthermore, resistance can be primary or acquired. Primary resistance to TDM-1 seems to be relatively infrequent, both in patients pre-treated with trastuzumab containing regimens and in naïve patients. Instead, acquired resistance is common: in fact, most of patients that initially benefit from the treatment eventually experience a PD [4,26].

### 2.1. Receptor Related Mechanisms of Resistance to T-DM1

Since HER2 expression is needed for T-DM1 activity, decreased levels or structural alterations of this receptor after prolonged exposure to the drug was supposed to be an important mechanism of resistance. In fact, under the selective pressure caused by T-DM1 exposure, cells with higher HER2 expression are more easily eliminated despite while those with lower expression of HER2 receptor are subjected to a more favourable clonal evolution [27]. Even previous treatment targeting HER2 play a role in HER2 expression, as emerged from the worse outcomes of patients treated with TDM-1 after the combination of trastuzumab and pertuzumab.

Even the heterogeneity in HER2 expression both intralesional and between primary tumor and distant metastases may be associated to inferior outcomes of TDM-1 therapy, as observed in the subgroup of patients enrolled in the KRISTINE trial with early locoregional progression, whose diseases were characterized by a more heterogeneous HER2 expression [28].

Additionally, the expression of truncated forms of HER2 receptor (p95HER2) and membrane-associated mucin (MUC4) are known to determine trastuzumab resistance, respectively through the absence of the subdomain IV and the masking of HER2, as well as the mutations of the kinase domain of HER2. 

All these mechanisms, even if not fully proved for TDM-1, are likely to contribute to the development of resistance to trastuzumab-based ADCs [29,30,31]. 

### 2.2. Intracellular Mechanisms of Resistance to T-DM1

The intracellular concentration of T-DM1 and its subsequent cytotoxic activity depend on the endocytic uptake of this agent.

In a preclinical model, TDM-1 resistant cells showed to internalize HER2 receptor in vesicles marked with of caveolin-1 (CAV1), the major structural protein of cholesterol-rich *caveolae,* required for endocytic transport. CAV1 overexpression demonstrated to negatively correlate with membrane HER2 expression and to affect trastuzumab-tumor binding; this was associated to reduced lysosomal colocalization and a consequent decreased sensitivity to TDM-1. In fact, in a preclinical model, the colocalization of CAV1 and T-DM1 was demonstrated to correlate with a decreased response to T-DM1, assuming CAV1 as a potential predictive biomarker of resistance to T-DM1 or other non-cleavable linker ADCs [32,33,34,35].

On the contrary, the overexpression of endophilin A2 (Endo A2), another protein involved in clathrin-independent endocytosis, seems to be associated to poor prognosis but to confer an increased sensitivity to trastuzumab and TDM-1 [36]. 

HER2 receptor, once internalized, is characterized by and extremely rapid recycling, which may affect the TDM-1 efficacy by reducing the amount of ADCs driven to lysosomes and subsequently released in the cytoplasm [37]. A recent preclinical study supported this hypothesis, correlating the HER2 recycling rate to lower intracellular TDM-1 levels [38].

Moreover, HER2 receptors undergo to proteolytic degradation in lysosome or proteasome. In fact, the inhibitors of the chaperone Heat Shock Protein 90 (HSP90), like 17-allyl-aminodemethoxy-geldanamycin (17-AAG), when combined to trastuzumab, demonstrated to down-regulate the HER2 expression by increasing its degradation. Hence, a faster or more efficacy elimination of HER2 receptor may affect HER2 expression rates and eventually the sensitivity to T-DM1 [39].

Furthermore, as the cleavage of the linker and the subsequent release of TDM-1 metabolites require the lysosome activity, an increased lysosome pH or a defective lysosome-to-cytosol transport activity may be involved in acquired resistance. For example, the loss or the silencing of *LSC46A3*, a gene encoding for a membrane protein involved in the transport of TDM-1 metabolites outside the lysosomes, was demonstrated to be associated to acquired resistance to TDM-1 in in vitro models [40]. 

As demonstrated for many chemotherapeutical agents, also mertansine metabolites are a substrate of p-Glycoprotein (MDR1), an active efflux transporter whose higher expression or activity is associated to poor response to chemotherapy, because of the lower intracellular concentrations of drugs. In fact, in vitro inhibition of MDR1 resulted in reversal of resistance to TDM-1 [40]. 

The activation of downstream pathways, such as PI3K/AKT, is a well-known mechanism of resistance to trastuzumab and is mainly related to *PIK3CA* mutations and loss of *PTEN*. An exploratory analysis from EMILIA trial evaluated the correlations between PTEN protein expression or *PIK3CA* mutations and TDM-1 efficacy, confirming that previous trastuzumab-based treatments enhanced the signalling throw this pathway (*PIK3CA* mutations were associated to worse outcomes in patients from control arm receiving lapatinib plus capecitabine), but that the benefit derived from TDM-1 was still substantial (T-DM1 appeared to be effective in both *PI3KCA*-mutated and wild-type tumors). 

Furthermore, also the loss of *PTEN*, present in almost 50% of mBC, contributes to the resistance phenotype and seems to be associated to worse outcomes of trastuzumab-based regimens, including TDM-1 [40].

Moreover, the PI3K/AKT pathway can be activated also by HER2/HER3 heterodimerization; in fact, higher expressions of HER3 receptor and its ligand, neuregulin (NRG1), were observed in TDM-1 resistant cells [41].

Lastly, defects in cyclin B1, a key factor for T-DM1-induced cell-cycle arrest during the G2/M transition, were observed in cells resistant to TDM-1, while sensitive cells showed higher levels of this protein.

Other potential mechanisms and pathways potentially involved in resistance to T-DM1 and not fully evaluated comprise: EGFR overexpression, STAT3 activation, ROR2 overexpression, altered forms of tubulin βIII and mutations of tubulins in general [40].

## 3. Emerging Mechanisms of Resistance to T-DXd: The DAISY Trial

T-DXd has recently emerged as a novel option in the management of patients with HER2-positive mBC, in second line setting, and with HER2low mBC progressing on at least one line of chemotherapy [18,42]. However, many aspects about T-DXd mechanism of action and resistance remain still unclear. First evidences came from results of DAISY trial [43].

The DAISY trial was the first study aiming at investigating the T-DXd mechanisms of action and resistance; it was a phase II, multicenter, open-label trial (NCT04132960) enrolling patients with HER2 positive (*n* = 72), HER2low (*n* = 74) or HER2-zero (*n* = 40) mBC, grouped in three parallel cohorts. T-DXd was administered at the dosage of 5.4 mg/kg q3w upon PD or unacceptable toxicity. The primary endpoint was the best objective response (BOR); secondary endpoints included PFS and duration of response (DOR) for patients presenting an objective response evaluated on the Full Analysis Set (FAS) and per cohort; the following were evaluated as exploratory objectives: T-DXd distribution, modulation of immune cells by T-DXd, dynamic of HER2 expression before and after T-DXd administration, development of predictors of response, identification of mechanisms of primary and secondary resistance. Therefore, in order to identify potential mechanisms of resistance to T-DXd, tumor biopsies were assessed at baseline and at progression, performing a whole exome sequencing (WES) analysis [43].

Most of patients were heavily pre-treated: 53% of them had received 5 or more lines of therapy in metastatic setting. BOR rates were 71%, 37.5% and 30% in the three cohorts, respectively. After a median follow-up of 15.6 months, median PFSs were respectively 11.1, 6.7 and 4.2 months [43].

A different distribution of T-DXd was observed between the cohorts: HER2 IHC 0 samples showed a lower T-DXd distribution than HER2-positive samples (*p* = 0.05). Furthermore, HER2 overexpressing areas showed an enrichment of serotonin and G-protein coupled receptor signalling after T-DXd administration; conversely, interferon alpha pathway was enriched in HER2-negative areas [43].

As far as potential mechanisms of resistance to T-DXd, driver alterations were identified in at least 3% of samples and those founded in baseline samples were associated with upfront resistance. However, 6% (5/88) of patients presented an *ERBB2* hemizygous deletion at baseline and four of them did not respond to T-DXd.

IHC for HER2 was assessed both in baseline and PD samples, and a decrease of HER2 expression at PD, as secondary resistance mechanism, was found in 65% (13/20) of patients. 

Genomic alterations were assessed by WES in 20 PD samples, half of whom were matched with baseline biopsies. Interestingly, mutations of *SLX4* gene, encoding a DNA-repair protein which regulate the endonucleases, were detected in 20% (4/20) of PD biopsies; half of them were acquired and the other 2 missed the matched baseline samples to perform the WES [44].

According to this evidence, BC cell lines depleted for *SLX4* were treated with DXd for 5 days and, interestingly, a higher quantity of DXd was required to kill them. Therefore, *SLX4* loss of function mutations could mediate resistance to the anti-topoisomerase 1 (TOP1) activity of the payload (deruxtecan) [43].

In conclusion, the decrease of HER2 expression and *SLX4* loss of function mutations represent the first evidences of mechanisms of resistance to T-DXd.

## 4. Novel Combination Strategies to Overcome or Prevent Resistance

Many strategies are under evaluation to overcome or prevent resistances to anti-HER2 ADCs and increase clinical outcomes in patients progressing on these agents; the most promising approaches include combinations of ADCs with TKI, statins, immune checkpoint inhibitors (ICIs) and synthetic DNA-damaging agents (Table 1, Figure 1).

**Figure 1 cancers-15-01130-f001:**
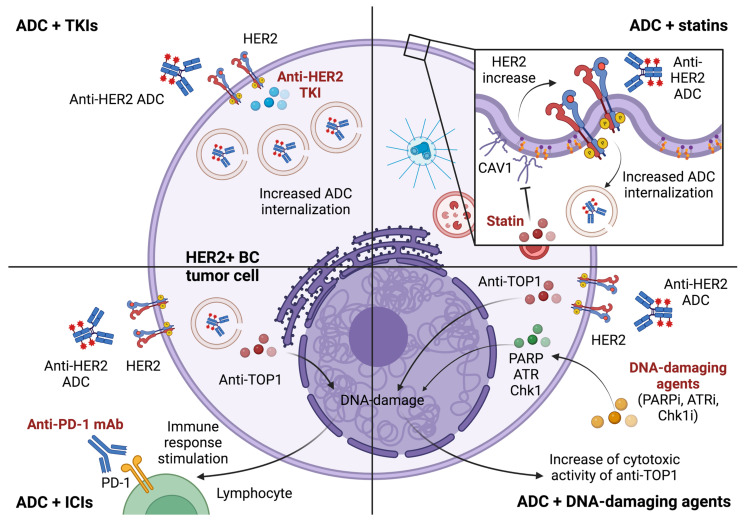
Combination approaches under evaluation to overcome resistance to anti-HER2 antibody-drug conjugates. Keys: ADC, antibody-drug conjugate; ATRi, ataxia telangiectasia and Rad3-related protein inhibitor; BC, breast cancer; Chk1i, checkpoint kinase 1 inhibitor; HER2, human epidermal growth factor receptor 2; ICI, immune-checkpoint inhibitor; mAb, monoclonal antibody; PARPi, PolyADP-Ribose Polymerase-1 inhibitor; PD-1, programmed death 1 protein; TKI, tyrosine kinase inhibitor; TOP1, topoisomerase 1. Created with BioRender.com.

### 4.1. Tyrosine Kinase Inhibitors and ADCs

In order to overcome resistances related to the decrease of HER2 expression or HER2 mutations [41,45], and, subsequently, to modulate the internalization of HER2 receptor, there is a rational to combine drugs acting on the extra- and intracellular domains of HER2 receptor, respectively mAbs/ADCs and TKIs.

Among the TKIs, the irreversible HER2 inhibitor neratinib showed good results in patients refractory to trastuzumab therapy in both the adjuvant [46] and metastatic settings in a phase II trial [47]. This led to the evaluation of neratinib plus T-DM1 in a phase 1b study in patients pre-treated with trastuzumab, pertuzumab and taxanes [48]. Promising signals of activity emerged. Furthermore, on circulating free DNA (cfDNA)-based baseline analyses, the decrease of HER2 levels and the occurrence of HER2 truncated forms (p95HER2) emerged as the main mechanism of resistance to dual blockade with trastuzumab and pertuzumab.

Moreover, since the combination of irreversible TKIs and ADCs has shown to increase the internalisation of the receptor and the subsequent uptake of the ADCs’ payload, the synergistic combination of a TKI with T-DXd may facilitate drug internalisation and overcome resistances related to the decrease of HER2 expression. To date, several studies are ongoing (Table 1) [49].

### 4.2. Statins and ADCs

As previously described, the alteration of T-DM1 internalization through the increase of CAV1 may represent a possible mechanism of resistance to T-DM1 [33]. According to preclinical evidences demonstrating the role of statins in CAV1 inhibition, their potential role in the subsequent prevention of HER2 internalization was evaluated; in fact, a recent in vitro study demonstrated a significant correlation between lovastatin intake and increased cell plasma membrane-bound HER2, as well as an improved sensitivity to trastuzumab in HER2-positive gastric cancer [50]. In order to investigate whether these results might be observed in the clinic, retrospective analyses has been recently conducted in a cohort of 164 patients with HER2-positive mBC who received T-DM1; interesting results has been reported with a median PFS of 14 months (95% confidence interval [CI], 3.5–24 months) in patients using statin compared to 5.4 months (95%CI, 3.9–7.0 months) in those who had no record of statin use (*p* = 0.1). In summary, these findings make statins as a potential therapeutic partner for anti-HER2 ADCs. Further studies are needed to better define the role of this association also in HER2-low setting [51]. 

### 4.3. Immune-Checkpoint Inhibitors and ADCs

The immune-stimulating effect related to the payload cytotoxicity led to investigate the combinations of immunotherapy and anti-HER2 ADCs. In particular, the combination of T-DM1 with ICIs showed greater signals of efficacy than T-DM1 monotherapy in patients pre-treated with trastuzumab, suggesting a potential enhancement of the immune system by conjugated maytansanoid [52].

Anti-TOP1 payloads, such as deruxtecan or govitecan, could stimulate both innate and adaptive immune response because of their DNA damage mediated effect; different mechanisms have been described, including cyclic GMP-AMP synthase (cGAS)/STING signalling pathway activation [53], DNA damage-induced antigen presentation via tumor cell MHC class I molecules [54], and DNA damage-induced release of tumor microvescicles [55]. Specifically, T-DXd demonstrated to increase immune-cells and to enhance PD-L1 and MHC class I expression on tumor cells in mouse models [56]; however slight results emerged in human cell lines affected by HER2-positive gastric cancer, where T-DXd demonstrated to provide only a modest upregulation of MHC I and CXCL9/10/11 [57]. According to this biological rationale, many trials are investigating the combination of T-DXd with ICIs (Table 1).

### 4.4. DNA-Damaging Agents and ADCs

Another possibility to enhance the cytotoxic effect of anti-TOP1 payloads may be the combination with other drugs sensitizing cells to DNA-damaging, such as PolyADP-Ribose Polymerase-1 (PARP) inhibitors [58], which demonstrated to halt DNA stabilization with a potential synergic role. Some investigation trials are ongoing (Table 1).

Other combination strategies may involve the inhibition of ATM–Chk2 and ATR–Chk1 pathways, two kinase signalling cascades activated by DNA double-strand and single-strand breaks, respectively [59]. In recent studies conducted in in vitro and xenograft models [60], the inhibition of ATR and Chk1 demonstrated to sensitize cancer cells to TOP1 inhibitors; according to these biological evidences many trials are investigating the combination of antibodies conjugated with anti-TOP1 and Chk1 or ATR inhibitors (Table 1).

## 5. Next Generation ADCs to Overcome or Prevent Resistances

In order to further improve clinical outcomes in patients with HER2-positive or HER2low mBC, many next generation ADCs, composed by different target and payloads, are currently under development (Table 1).

Trastuzumab duocarmazine (SYD985) demonstrated highly in vitro effectiveness in killing cancer cells expressing different levels of HER2, through a bystander killing of HER2-negative cells, and a superiority to T-DM1, with a 10–70-fold increase in potency in BC patient-derived xenografts models [61]. Furthermore, the objective response rate of 40% emerged in heavily treated patients with HER2-positive mBC led to Trastuzumab duocarmazine fast-designation approval from FDA in January 2018 [62]; results from the cohort of patients with HER2low (*n* = 47) were also encouraging, with an objective response achieved in 28% of patients with hormone receptor-positive mBC. The phase 3 trial TULIP evaluated the efficacy Trastuzumab duocarmazine versus physician’s choice chemotherapy in patients with HER2-positive mBC pre-treated with at least two anti-HER2 regimens, including T-DM1; despite a benefit in median PFS (7.0 vs. 4.9 months; HR 0.64, 95% CI: 0.49–0.84; *p* = 0.002), many safety concerns emerged and 35% (vs. 10%) of patients discontinued the treatment because of adverse events [63].

Disitamab vedotin showed promising results in HER2low setting and is currently being investigated in a HER2low mBC population versus chemotherapy in a phase III trial (NCT04400695) [64].

Furthermore, in preclinical studies, Zanidatamab zodovotin, a novel bispecific-ADC targeting 2 different HER2 epitopes and conjugated to an anti-microtubule agent by a cleavable linker, demonstrated to determine HER2 receptor clustering, rapid receptor internalization, inhibition of recycling, and lysosomal degradation [65]. These mechanisms could be useful in order to overcome acquired mechanisms of resistance to anti-HER2 ADCs. Zanidatamab zodovotin is currently being studied in a phase I trial (NCT03821233) and preliminary results demonstrated encouraging responses in heavily pretreated patients with HER2-positive solid cancers [66].

As far as targets other than HER2, HER3 overexpression seems to be associated to resistance to anti-HER2 agents (both trastuzumab and T-DM1) [41]; in fact, HER3 inhibition demonstrated to overcome the acquired trastuzumab resistance in HER2-positive gastric cancer-derived xenograft [67]. According to these data, there is a strong preclinical rationale to evaluate anti-HER3 agents, and in particular anti-HER3 ADCs, such as Patritumab deruxtecan, in patients progressing to anti-HER2 ADCs with HER3-high mBC. Interestingly, in a small subgroup of 14 heavily pre-treated patients with HER2-positive HER3-high mBC, Patritumab deruxtecan showed important signals of activity, with an objective response rate of 43% and a disease control rate of 93% [68]. An ongoing single arm phase 2 clinical trial (NCT04699630) aims at recruiting 120 pre-treated patients with HER2-positive HER3-high mBC and at evaluating the activity of this promising ADC [69].

## 6. Current Issues and Future Challenges

As stated, many novel agents, including ADCs, and combination approaches are emerging as promising strategies for the treatment of patients with HER2-positive or HER2low mBC. For example, T-DXs showed an ever-progressive improvement of clinical outcomes in third- and second-line settings, in DESTINY-Breast02 and 03 clinical trials, respectively [70].

The positioning in ever-earlier lines of novel ADCs raises the issue about the efficacy of older agents in later lines, after a PD on more effective therapies. For example, the efficacy of T-DM1, pertuzumab or lapatinib after T-DXd is not fully understood and should be prospectively evaluated, also considering the rate of intrinsic subtype switch of HER2-positive mBC under the selective pressure of anti-HER2 therapies. For example, the dual HER2 blockade with neoadjuvant trastuzumab, lapatinib and paclitaxel in patients with PAM50 HER2-enriched early BC demonstrated to induce low-proliferative Luminal A phenotype [71]. However, in the last update of DESTINY-Breast03 clinical trial, promising signals in terms of PFS to the subsequent therapy (PFS2) emerged (40 vs. 26 months; hazard ratio [HR]: 0.47; 95%CI: 0.35–0.62), highlighting that T-DXd may not negatively impact on the efficacy of subsequent treatments (35% of patients from T-DXd arm received T-DM1) [70]. 

In order to optimize the repositioning of older agents, and to prioritize the development of all these emerging agents and strategies, a huge effort to identify biomarkers of response and resistance is required. For example, loss of function *neurofibromatosis 1* (*NF1*) mutations, which account for about 8% of mBC, are associated with microtubule dysfunction; in fact, chromosome alignment defects and multipolar spindle formations were founded in *NF1*-mutant cellular models of HER2-positive BC [72,73]. According to this preclinical evidence, it has been demonstrated that response to T-DM1 was significantly increased in *NF1* knockout models (IC50 ~0.3 vs. 1.6 μg/mL in *NF1*-WT); this sensitization was not observed with other ADCs and was reproducible with mertansine alone, suggesting a pharmacologically relevant NF1 activity on microtubules. Therefore, there is as strong preclinical rationale to investigate T-DM1 and ADCs with anti-microtubule as payload in patients with HER2-positive *NF1*-mutant BC progressing on T-DXd.

Importantly, as far as resistance, the mere transfer of resistance mechanisms and biomarkers of older agents (e.g., trastuzumab or pertuzumab) to novel drugs (e.g., T-DXd) should be avoided. In fact, for example, the downstream *MAPK* or *PI3K* alterations emerged as mechanisms of resistance to trastuzumab and pertuzumab, but not to T-DM1 and T-DXd [74]. Therefore, a continuous effort of combination of preclinical data and clinical evidences is needed in order to identify further intrinsic and extrinsic mechanisms of resistance, and to better understand the interactions between overexpressed tumor cell pathways and tumor microenvironment [4].

Once identified in preclinical and clinical setting, biomarkers of resistance should be assessed in terms of analytical validity and draggability. In order to better depict the spatial heterogeneity of drug-resistance in metastatic setting, samples should be acquired by both tissue and liquid biopsies, preferably with a multiple-time point and longitudinal approach. Furthermore, prospective studies are needed for the assessment of a biomarker clinical validity, and, finally, the demonstration its clinical utility requires clinical trials using the biomarker status for patient selection or patient stratification [75].

However, the high speed of new drug development does not allow all these questions to be answered with *post-hoc* prospective validations of biomarkers. Therefore, there is an urgent need to further enhance the translational backbone of clinical trials and to share the results of biomarker analyses with the same speed as drug development [76]. In this regard, a major effort to identify and validate surrogate endpoints to accelerate not only drug approval but also biomarker analysis for patient selection is required.

## 7. Conclusions

In conclusion, despite the advent of anti-HER2 ADCs represented an important therapeutic breakthrough in patients with HER2-positive and HER2low mBC, their benefit is limited because of the occurrence of acquired and *de novo* resistances. Therefore, a further effort to overcome resistance through novel strategies and approaches is required. DAISY trial paved the way in the study of T-DXd mechanisms of action and resistance, providing encouraging but still inconclusive results. In addition, intriguing preclinical data led to several clinical studies evaluating possible therapeutic combinations to enhance ADC efficacy, and overcome or prevent resistances. Finally, novel emerging ADCs, such as Trastuzumab duocarmazine, Disitamab vedotin and Zanidatamab zodovotin, demonstrated promising signals of activity in both HER2-positive and HER2low setting.

The next challenge will be the definition of a treatment algorithm including all these emerging agents and combination approaches. In this regard, there is an urgent need to identify the most promising biomarkers of resistance and response to anti-HER2 ADCs, by longitudinal sampling of tissue and liquid biopsies, and, subsequently, to validate their clinical utility in prospective clinical trials.

## Figures and Tables

**Table 1 cancers-15-01130-t001:** Ongoing clinical trials evaluating combinations of antibody-drug conjugates to overcome or prevent resistances in HER2-positive breast cancer.

Treatment	NCT Number	Ph.	Patients	Study Design	Endpoints
**ADCs + TKIs**					
T-DM1 + lapatinib	NCT02073916	Ib	HER2+ mBC	T-DM1 + Lapatinib + nab-paclitaxel	Safety
T-DM1 + tucatinib	NCT01983501	I	HER2+ mBC	T-DM1 + Tucatinib	Safety
T-DM1 + neratinib	NCT02236000	Ib/II	HER2+ mBC	T-DM1 + Neratinib	Safety, ORR
T-DM1 + lapatinib	NCT02073487	II	HER2+ BC	Neoadjuvant T-DM1 + Lapatinib followed by Abraxane vs. Trastuzumab + Pertuzumab Followed by Paclitaxel	pCR, RCB
T-DM1 + neratinib	NCT05388149	II	MRD+ HER2+ BC	T-DM1 + Neratinib	ctDNA clearence
T-DXd + tucatinib	NCT04539938	II	HER2+ mBC	T-DXd + Tucatinib	ORR
T-DM1 + tucatinib	NCT03975647	III	HER2+ mBC	Tucatinib/Placebo + T-DM1	PFS, OS
T-DM1 + tucatinib	NCT04457596	III	High risk HER2+ BC	Tucatinib/ Placebo + T-DM1	iDFS
**ADCs + ICIs**					
T-DM1 + pembrolizumab	NCT03032107	Ib	HER2+ mBC	Pembrolizumab + T-DM1	Safety
T-DM1 + atezolizumab	NCT02605915	Ib	HER2+ BC	Atezolizumab + T-DM1 vs. (Atezolizumab + Trastuzumab + Pertuzumab ± Docetaxel)	Safety
T-DM1 + utomilumab	NCT03364348	Ib	HER2+ mBC	Utomilumab + T-DM1 vs. Utomilumab + Trastuzumab	Safety, ORR
T-DXd + Durvalumab	NCT04556773	Ib	HER2low mBC	T-DXd + Durvalumab	Safety, ORR
T-DXd + Nivolumab	NCT03523572	Ib	HER2+ mBC	T-DXd + Nivolumab	Safety, ORR
T-DXd + pembrolizumab	NCT04042701	Ib	mBC	T-DXd + Pembrolizumab	Safety, ORR
T-DM1 + atezolizumab	NCT02924883	II	HER2+ mBC	T-DM1 + Atezolizumab/placebo	PFS, OS, ORR
T-DM1 + atezolizumab	NCT04740918	III	HER2+ PDL1+ mBC	T-DM1 + Atezolizumab/placebo	PFS, OS
**ADCs + DNA-damaging agents**					
T-DXd + Ceralasertib	NCT04704661	I/Ib	HER2+ solid tumors	Ceralasertib + T-DXd	Safety

Keys: ADCs, antibody-drug conjugates; ctDNA, circulating tumor DNA; HER2, human epidermal growth factor receptor 2; ICI, immune checkpoint inhibitor; iDFS, invasive disease-free survival; mBC, metastatic breast cancer; ORR, overall response rate; OS, overall survival; pCR, pathological complete response; PD-L1, programmed death ligand 1; PFS, progression free survival; Ph, phase; T-DM1, trastuzumab emtansine; T-DXd: trastuzumab deruxtecan; TKIs: tyrosine kinase inhibitors.

## Data Availability

Not applicable.

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
