# Peer review of "Resistance to Antibody-Drug Conjugates Targeting HER2 in Breast Cancer: Molecular Landscape and Future Challenges"

_cancers, 2023, doi:10.3390/cancers15041130_

Round 1

Reviewer 1 Report (Previous Reviewer 2)

Comments on cancers-2187894

The present manuscript is a resubmitted manuscript of “cancers-2148606.” According to previous reports, the authors only addressed the introduction and some references. No mechanism is added, no proofreading is done, and no new information is added in some sections. The paragraph’s length is also not revised in the updated manuscript. Sentence-making is still poor in this manuscript.

Specific comments:

1.      Please add a figure of signaling involved in the resistance mechanism to T-DM1.

2.      Please add a figure of signaling involved in the resistance mechanism to T-DXd.

3.      According to this evidence anti-topoisomerase 1 (TOP1) activity of the payload (deruxtecan)..” Please add a reference here.

4.      Page 6, line 255: “Among the TKIs, the irreversible HER2 inhibitor neratinib showed good results…” Neratinib is not a novel tyrosine kinase inhibitor. What is novel in section 4.1?

5.      “a significant correlation between statin intake and increased cell plasma membrane-bound HER2, as well as an improved sensitivity to trastuzumab in HER2-positive gastric cancer.” Which class of statin was used?

6.      Section 4.2 contain very little and general information; please add more information about specific statin.

7.      Section 4. “Novel combination strategies to overcome or prevent resistance,” contain general information. This information is already present in the literature. What is novel in this section?

8.      What is the next-generation ADCs?

9.      The conclusion should be in a single paragraph.

10.  Authors are advised to proofread the manuscript to overcome grammatical mistakes.

11.  Authors are advised to revise headings and subheadings.

Author Response

  1. Please add a figure of signaling involved in the resistance mechanism to T-DM1. Thanks for the suggestion, as stated in the previous revision, we preferred not to include a figure describing the resistance mechanism to T-DM1 since it has been yet published by Hunter and colleagues (DOI: 10.1038/s41416-019-0635-y)

  1. Please add a figure of signaling involved in the resistance mechanism to T-DXd. Thanks for the suggestion, as stated in the previous revision, we preferred not to add a figure on resistance mechanisms to T-Dxd, because only the decrease of HER2 expression and SLX4 loss of function mutations were described as resistance mechanisms
  2. “According to this evidence… anti-topoisomerase 1 (TOP1) activity of the payload (deruxtecan)..” Please add a reference here. Thanks for the suggestion, we added this reference
  3. Page 6, line 255: “Among the TKIs, the irreversible HER2 inhibitor neratinib showed good results…” Neratinib is not a novel tyrosine kinase inhibitor. What is novel in section 4.1? Thanks for your question, as stated in the previous revision, in 4.1 section we tried to summarize the rationale behind the potential combinational strategy adding TKIs to ADCs, according to first data about the evaluation of the combination of Neratinib + T-DM1.

  1. “a significant correlation between statin intake and increased cell plasma membrane-bound HER2, as well as an improved sensitivity to trastuzumab in HER2-positive gastric cancer.” Which class of statin was used? Thanks for asking, according to the previous revision, we have just modified the manuscript adding the specific statin class used in the study (lovastatin). Are you sure that you’ve read the last version of the manuscript?
  2. Section 4.2 contain very little and general information; please add more information about specific statin. Thanks for your suggestion, as stated in the previous revision, unfortunately almost all evidences about this topic are preclinical, we tried to explain the rationale behind this potential strategy. Lovastatin has been used in the study we mentioned
  3. Section 4. “Novel combination strategies to overcome or prevent resistance,” contain general information. This information is already present in the literature. What is novel in this section? Thanks for asking, as stated in the previous revision, in this chapter we aim to summarize the main combination strategies which could represent an option as a potential way to overcome resistances or enhance ADC’s efficacy.
  4. What is the next-generation ADCs? Thanks for asking, this is the definition “By that time, the progressive transition from non-cleavable to cleavable linkers, the optimization of Abs structures and the improvement of DAR, leaded to the development of new generations of anti-HER2 ADCs in mBC” (pag. 3 lines 100-103)

  1. The conclusion should be in a single paragraph. Yes, it should be. As stated in the previous revision, we merged two of three paragraphs, according to instructions for authors “This section is mandatory, with one or two paragraphs to end the main text”.

  1. Authors are advised to proofread the manuscript to overcome grammatical mistakes. Thank for the suggestion, we corrected all mistakes.

  1. Authors are advised to revise headings and subheadings. Thank for the suggestion, we revised all headings and subheadings, and modified the first headings

Reviewer 2 Report (Previous Reviewer 1)

Please rewrite the following using in proper sentences, "

To date, many other anti-HER2 ADCs are being tested in clinical trials (Figure 1), including trastuzumab duocarmazine (SYD985), which is conjugated by a cleavable linker to the alkylating agent duocarmycin, with a DAR of 2.8:1; disitamab vedotin (RC48-ADC),   101 which conjugates the anti-HER2 mAb disitamab to the synthetic antineoplastic agent  monomethyl auristatin E (MMAE), an anti-microtubule agent, with a DAR of 4; XMT-1522  (TAK-522), a new generation ADC targeting different HER2 epitopes than trastuzumab,  whose development was recently halted due to safety issues; ARX788, a next-generation  site-specific anti-HER2 which is conjugated with amberstatine, an anti-microtubule agent,among others [20–24]."

Also, "mAbs incorporated in ADCs derive from three IgG isotypes (IgG1, IgG2 and IgG4) 803 and are linked to a precise number of payload molecules, depending on the linker and the 804 payloads characteristics [9]; the drug to antibody ratio (DAR) is usually between 2 and 8."

Also, please name the linkers. "Linkers affect all pharmacokinetic properties of ADCs and can be cleavable or non-806 cleavable; non-cleavable linkers are more stable and are cleaved only by the complete pro-807 teolytic degradation of the ADC in lysosomes [10]. Instead, cleavable linkers release the 808 payload in response to tumor-associated factors, such as acid pH or abundance of prote-809 olytic enzymes, allowing the diffusion of the payload through neighbouring cells not ex-810 pressing the target (bystander effect) [11]."

Author Response

Please rewrite the following using in proper sentences, "

To date, many other anti-HER2 ADCs are being tested in clinical trials (Figure 1), including trastuzumab duocarmazine (SYD985), which is conjugated by a cleavable linker to the alkylating agent duocarmycin, with a DAR of 2.8:1; disitamab vedotin (RC48-ADC), 101 which conjugates the anti-HER2 mAb disitamab to the synthetic antineoplastic agent    monomethyl auristatin E (MMAE), an anti-microtubule agent, with a DAR of 4; XMT-1522           (TAK-522), a new generation ADC targeting different HER2 epitopes than trastuzumab,          whose development was recently halted due to safety issues; ARX788, a next-generation site-specific anti-HER2 which is conjugated with amberstatine, an anti-microtubule agent, among others [20–24]."

Thanks for the suggestion, we emended the manuscript according to your advice.

“To date, many other clinical trials are testing novel anti-HER2 ADCs (Figure 1). They include: Trastuzumab Duocarmazine (SYD985), which is formed by the conjugation of trastuzumab to the alkylating agent duocarmycin mediated by a cleavable linker with a DAR of 2.8:1; Disitamab Vedotin (RC48-ADC), 101 which is characterised by the conjugation of the anti-HER2 mAb disitamab to the synthetic antineoplastic agent monomethyl auristatin E (MMAE), an anti-microtubule agent, with a DAR of 4; XMT-1522 (TAK-522), a new generation ADC targeting different HER2 epitopes than trastuzumab, whose development was recently halted due to safety issues; ARX788, a next-generation site-specific anti-HER2 which is conjugated with amberstatine, an anti-microtubule agent, among others [20–24].

Also, "mAbs incorporated in ADCs derive from three IgG isotypes (IgG1, IgG2 and IgG4) 803 and are linked to a precise number of payload molecules, depending on the linker and the 804 payloads characteristics [9]; the drug to antibody ratio (DAR) is usually between 2 and 8."

Thanks for the suggestion, we emended the manuscript according to your advice.

“mAbs incorporated in ADCs are mainly based on the igG1 isotype because of its higher immunogenic properties and easier production if compared to the other igG2 and igG4 subtypes [9]; the drug to antibody ratio (DAR) is usually between 2 and 8. ”   

Also, please name the linkers. "Linkers affect all pharmacokinetic properties of ADCs and can be cleavable or non-806 cleavable; non-cleavable linkers are more stable and are cleaved only by the complete pro-807 teolytic degradation of the ADC in lysosomes [10]. Instead, cleavable linkers release the 808 payload in response to tumor-associated factors, such as acid pH or abundance of prote-809 olytic enzymes, allowing the diffusion of the payload through neighbouring cells not ex-810 pressing the target (bystander effect) [11]."

Thanks for the suggestion, we emended the manuscript according to your advice

"Linkers affect all pharmacokinetic properties of ADCs and can be cleavable or non-806 cleavable; non-cleavable linkers are more stable and are cleaved only by the complete pro-807 teolytic degradation of the ADC in lysosomes [10]. Instead, cleavable linkers release the 808 payload in response to tumor-associated factors, such as acid pH (acid-labile linkers), reduction-oxidation conditions (disulfide-linkers), abundance of prote-olytic enzymes(protease-cleavable linkers), allowing the diffusion of the payload through neighbouring cells not ex-810 pressing the target (bystander effect) [11]."

This manuscript is a resubmission of an earlier submission. The following is a list of the peer review reports and author responses from that submission.

Round 1

Reviewer 1 Report

1. The manuscript reveals over 350 grammar errors with 13% plagiarism which requires atttention, and unacceptable for publication.

2. There are statements throughout the manuscript that require proper citations. For eample, "Furthermore, HER2 overexpressing areas showed an enrichment of serotonin and G-protein 224 coupled receptor signalling after T-DXd administration; conversely, interferon alpha pathway was enriched in HER2-negative areas. (ref??)".

3. Most of the manuscript display statements without going into the depth and mechanism(s) of action. Why do they work or not work?

4. Hunter et al. (British Journal of Cancer (2020) 122:603–612; https://doi.org/10.1038/s41416-019-0635-y) provide a comprehensive review of the Mechanisms of resistance to trastuzumab emtansine (T-DM1) in HER2-positive breast cancer. This study was not cited.

5. The challenges and failures of breast cancer treatments must be articulated. For example, many candidate pathways can be exploited in breast cancer patients depending on their unique tumor microenvironment and genetic profile. While these candidate drugs may all have promising benefits to extend survivorship, realistically, the efficacy of these additional inhibitors depends on the unique profile of the disease and whether the cells express the molecular targets upon which candidate inhibitors can act. Furthermore, it has been well established that the tumor microenvironment or the tumor itself has mechanisms to compensate for the inhibition of one pathway by upregulating others. Therefore, designing multimodal therapies must not only target a particular molecular pathway facilitating tumor growth and survival, but target the compensating pathways upregulated by the initial targeted approach.

6. Resistance to chemotherapy can present itself in two forms: intrinsic resistance, where cancer cells are inherently resistant to chemotherapy in patients receiving chemotherapy for the first time, and acquired resistance, where cancer cells become desensitized to chemotherapy because of continued exposure to this treatment. Chemoresistance is responsible for 90% of therapy failure in the treatment of metastatic cancer, including metastatic breast cancer (Longley DB, Johnston PG. Molecular mechanisms of drug resistance. J Pathol. 2005;205(2):275-292).

7. The authors need to articulate the overview of the treatment options for metastatic breast cancer. Depending on the location and
size of metastases, local or systemic therapy may be chosen. Therapy will also depend on the cancer cell subtypes present in the patient, as well as the patient’s menopausal status.

8. The authors need to cite properly their statements.

Reviewer 2 Report

Comments on cancers-2148606

In this study, the author has studied “Resistance to Antibody-Drug Conjugates targeting HER2 in Breast Cancer: molecular landscape and future challenges.” A lot of studies have already been carried out on a similar topic, and comprehensive data is available in the literature. The present manuscript lacks novelty and the main objective. It also did not explain any new idea and contained repeated basic information that is already present in the literature. Sentence-making is very poor in this manuscript. The English language used in the manuscript needs major improvements as some punctuation, and grammatical mistakes are present. Experimental designs required more clarity. Moreover, research models are not discussed in an understandable manner. Repetition of lines is common in the manuscript, which reflects that the author needs a more comprehensive way of thinking. Each paragraph should be about 10-15 lines long. Important references are missing. It looks like the authors are trying to write an essay instead a scientific manuscript. Table 1 is missing. It is obvious that the quality of the manuscript does not fulfill the standards of the journal. Therefore, it should be rejected in its present form.

Specific comments:

1.      Page 1, line 34-35: “Promising approaches include combinations with tyrosine kinase inhibitors, statins, immune checkpoint inhibitors and synthetic DNA-damaging agents, among others.” Please revise the sentence.

2.      The Abstract needs to be critically revised; please limit the background knowledge to a few sentences and add a clear objective of the study.

3.      Please add more strong keywords.

4.      The first heading should be an introduction.

5.      Page 1, line 42-43: “Breast cancer (BC) represents the leading cancer for incidence and mortality in women [1].” Please also add the latest epidemiology of breast cancer.

6.      Page 2, line 57-59: “ADCs are an evolving class of anticancer drugs… these elements and their specific features.” Please add a reference here.

7.      Page 2, line 63-67: “Linkers affect all pharmacokinetic properties… not expressing the target (bystander effect).” Please add a reference here.

8.      Page 2, line 86: “amplification at in situ hybridization (ISH) test.” In situ should be italic (in situ).

9.      Page 3: The whole section 1 looks general. Authors are advised to revise section 1 carefully and add relevant data to support the problem statement and make a connection between each paragraph. The length of the paragraphs is very ordinary, each paragraph should be 10-15 lines long. Important references are missing in this section. Please add relevant literature. Overall, an introduction needs a major revision.

10.  Page 3: What is the research gap and novelty of the present study?

11.  Please add a figure of signaling involved in the resistance mechanism to T-DM1.

12.  Page 3, line 108-111: “Therefore, the mechanisms of resistance…mere sum of their 111 components.” Please add a reference here.

13.  Page 3, line 113-115: “Mechanisms of resistance to T-DM1 can be classified… plex regulation of the intracellular signalling pathways.” Please add a reference here.

14.  Page 3, line 122-129: “Since HER2 expression is needed for…after the combination of trastuzumab and pertuzumab.” Please add a reference here.

15.  Please add a figure of signaling which involved in the mechanism of resistance to T-DXd

16.  Page 5, line 205-217: “The DAISY trial was the first study aiming at…forming a whole exome sequencing (WES) analysis.” Please add a reference here.

17.  Page 5, line 218-220: “Most of patients were heavily pre-treated:…PFSs were respectively 11.1, 6.7 and 4.2 months.” Please add a reference here.

18.  Page 5, line 222-226: “A different distribution of T-DXd was observed… pathway was enriched in HER2-negative areas.” Please add a reference here.

19.  Where is Table 1?

20.  Page 6, line 255: “Among the TKIs, the irreversible HER2 inhibitor neratinib showed good results…” Neratinib is not a novel tyrosine kinase inhibitor. What is novel in section 4.1?

21.  Page 7, line 280-282: “a significant correlation between statin intake and increased cell plasma membrane-bound HER2, as well as an improved sensitivity to trastuzumab in HER2-positive gastric cancer.” Which class of statin was used?

22.  Section 4.2 contain very little and general information; please add more information about specific statin.

23.  Section 4. “Novel combination strategies to overcome or prevent resistance,” contain general information. This information is already present in the literature. What is novel in this section?

24.  What is the next-generation ADCs?

25.  The conclusion should be in a single paragraph.

26.  Please add a list of abbreviations.

27.  Authors are advised to proofread the manuscript to overcome grammatical mistakes.

28.  Authors are advised to revise headings and subheadings.